

# Critical analysis of multiple reentrant localization in an antiferromagnetic helix with transverse electric field: Hopping dimerization-free scenario

Sudin Ganguly[1★], Sourav Chattopadhyay[2†], Kallol Mondal[3,4‡] and Santanu K. Maiti[5∘]

**1** Department of Physics, School of Applied Sciences,
University of Science and Technology Meghalaya, Ri-Bhoi-793101, India
**2** Department of Physics, The ICFAI University Tripura,
Kamalghat, West Tripura-799210, India
**3** School of Physical Sciences, National Institute of Science Education and Research,
Bhubaneswar, Jatni, Odisha-752050, India
**4** Homi Bhabha National Institute, Training School Complex,
Anushaktinagar, Mumbai-400094, India
**5** Physics and Applied Mathematics Unit, Indian Statistical Institute,
203 Barrackpore Trunk Road, Kolkata-700108, India

★ sudinganguly@gmail.com , † sourav.nbp@gmail.com ,
‡ kallolsankarmondal@gmail.com , ∘ santanu.maiti@isical.ac.in

## Abstract

Reentrant localization (RL), a recently prominent phenomenon, traditionally links to the interplay of staggered correlated disorder and hopping dimerization, as indicated by prior research. Contrary to this paradigm, our present study demonstrates that hopping dimerization is not a pivotal factor in realizing RL. Considering a helical magnetic system with antiferromagnetic ordering, we uncover spin-dependent RL at multiple energy regions, in the *absence* of hopping dimerization. This phenomenon persists even in the thermodynamic limit. The correlated disorder in the form of Aubry-André-Harper model is introduced by applying a transverse electric field to the helical system, circumventing the use of traditional substitutional disorder. We conduct a finite-size scaling analysis on the observed reentrant phases to identify critical points, determine associated critical exponents, and examine the scaling behavior linked to localization transitions. Additionally, we explore the parameter space to identify the conditions under which the reentrant phases occur. Described within a tight-binding framework, present work provides a novel outlook on RL, highlighting the crucial role of electric field, antiferromagnetic ordering, and the helicity of the geometry. Potential applications and experimental realizations of RL phenomena are also explored.

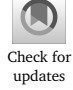

# 1   Introduction

Disorder tends to drive systems toward localization of electronic states. Despite this, the single-particle wave function displays significant distinctions in behavior when exposed to uncorrelated (random) disorder compared to correlated (quasiperiodic) one. In the presence of random disorder, Anderson localization [1, 2] takes place. The scaling theory [3] of Anderson localization predicts that single-particle states in one- and two-dimensional systems will exhibit exponential spatial localization, even when subjected to extremely weak disorder. Consequently, this leads to the absence of a single-particle mobility edge [4] (SPME). However, an energy-dependent mobility edge can exist in three-dimensional systems. In the context of Anderson localization, extensive investigations have been conducted on numerous fascinating systems across various branches of physics [2, 5–10].

In contrast to the uncorrelated disorder, correlated disorder provides the advantage of a sharply-defined critical point for the extended-localized phase transition [11–13] in Aubry-André-Harper (AAH) model [11, 14], as well as exhibiting fractal eigenmodes [15, 16] in Fibonacci model, and critical behavior [12, 17] in low-dimensional cases. Among the variety of quasiperiodic models [11–13, 15–22], the AAH model stands out as the most widely recognized and versatile example. The AAH model, akin to Anderson localization, lacks SPME due to the presence of a distinctly defined critical point characterizing the extended-localized phase transition [11–13]. Nonetheless, researchers have explored various generalizations of the standard AAH model to overcome this limitation. These extensions encompass diverse features, such as exponential short-range hopping [23], flatband networks [24], higher dimensions [25], power-law hopping [26], flux-dependent hopping [27], and nonequilibrium generalized AAH models [28], among others. Furthermore, the feasibility of AAH systems has been demonstrated through experimental realizations utilizing cold atoms and optical waveguides [29, 30]. These experimental implementations provide crucial platforms for studying the behavior of AAH models in controlled settings, offering valuable opportunities to explore and validate theoretical predictions in the realm of quantum simulation and condensed matter physics.

Conventionally, it is known that once a state is localized, it continues to remain localized even when the disorder strength is increased. However, Hiramoto and Kohmoto [31] demonstrated that under specific conditions, this behavior can deviate from the traditional understanding. Recent studies have further expanded on this phenomenon, revealing the possibility of a 'band-selective' localization-delocalization transition in various systems. Notably, such transitions have been observed in a one-dimensional tight-binding chain [32], validated using cavity-polariton devices [32], and in a spin chain with antiferromagnetic nearest-neighbor (NN) coupling subjected to an interpolating Aubry-André-Fibonacci on-site potential modulation [33]. Additionally, a similar transition has been demonstrated in a 1D chain under the influence of the off-diagonal interpolating Aubry-André-Fibonacci model [34]. Another

interesting work has been done by Roy and co-workers, where they have shown localization-to-delocalization transition, considering the interplay among the hopping dimerization and staggered on-site energies [35] implemented by AAH model. The reappearance of delocalized states (all and/or a certain number of all states) from the localized ones is referred to as 'reentrant localization' phenomenon, a term first introduced by Hiramoto and Kohmoto [31]. Although research on staggered AAH systems is relatively limited [36–42], these investigations highlight a notable distinction. Studies such as Refs. [31–34] achieve RL behavior by varying the interpolating parameter, which essentially changes the type of uncorrelated disorder, whereas others, including Refs. [35–42], focus on fixed types of uncorrelated disorder within the AAH model framework. A key observation from the latter group of studies is the suggestion that hopping dimerization is a primary requirement for RL. It essentially triggers us a fundamental question that *is it possible to realize RL behavior within a system governed by a specific type of uncorrelated disorder, but in the 'absence' of hopping dimerization?* This is the central question guiding our current investigation.

Here we propose a single-stranded antiferromagnetic helical system (see Fig. 1) in which neighboring magnetic moments are aligned along $\pm z$ directions (our chosen spin quantization axes). The helix is subjected to an external electric field, perpendicular to the helix axis. The motivations behind the consideration of such a system are many-fold. First, due to the helical geometry, site energies are modulated in the well-known AAH form in presence of transverse electric field [43–49]. Thus, the helix maps to an AAH system without imposing any substitutional disorder. In the absence of helicity or electric field, site energy modulation is no longer obtained. Second, due to such an arrangement of magnetic moments, the staggered condition in site energies is easily satisfied. Third, the observation of spin-specific phenomenon in a magnetic system with zero net magnetization is another challenge, and in our case we can successfully avail it imposing the interplay between the helicity and electric field. The localization phenomena are examined by inspecting various aspects, such as the single-particle energy spectrum, inverse participation ratio (IPR), participation ratio (PR), and other relevant measures. We determine the critical points and corresponding critical exponents for the different reentrant phases by defining an appropriate order parameter, following a theory analogous to thermal phase transition [51, 52]. We then verify these results using finite-size scaling theory.

The new and essential findings of our work are: (i) occurrence of spin-dependent RL in the absence of any hopping dimerization, (ii) observation of RL at multiple energies, and (iii) persistence of RL phenomena even in the thermodynamic limit.

## 2 System and theoretical framework

The schematic diagram of the proposed setup is illustrated in Fig. 1. It depicts a right-handed antiferromagnetic helix (AFH) comprising $N$ magnetic sites, where the magnetic moments of successive sites are aligned in opposite directions ($\pm z$). An external electric field of magnitude $E_g$ is applied perpendicular to the helix axis.

The AFH system in the presence of an electric field is described within the tight-binding framework and the corresponding Hamiltonian is

$$H = \sum_{n=1}^{N} \boldsymbol{c}_n^{\dagger} (\boldsymbol{\epsilon}_n - \hbar_n \cdot \boldsymbol{\sigma}) \boldsymbol{c}_n + \sum_{n=1}^{N} \sum_{m=1}^{N-n} \left( \boldsymbol{c}_n^{\dagger} \boldsymbol{t}_m \boldsymbol{c}_{n+m} + h.c. \right), \tag{1}$$

where, $\boldsymbol{c}_n^{\dagger}$, $\boldsymbol{c}_n$, $\boldsymbol{\epsilon}_n$, $\boldsymbol{t}_m$ read as

$$\boldsymbol{c}_n^{\dagger} = \begin{pmatrix} c_{n\uparrow}^{\dagger} & c_{n\downarrow}^{\dagger} \end{pmatrix}, \qquad \boldsymbol{c}_n = \begin{pmatrix} c_{n\uparrow} \\ c_{n\downarrow} \end{pmatrix}, \qquad \boldsymbol{\epsilon}_n = \text{diag}\left( \epsilon_n, \epsilon_n \right), \qquad \boldsymbol{t}_m = \text{diag}\left( t_m, t_m \right). \tag{2}$$

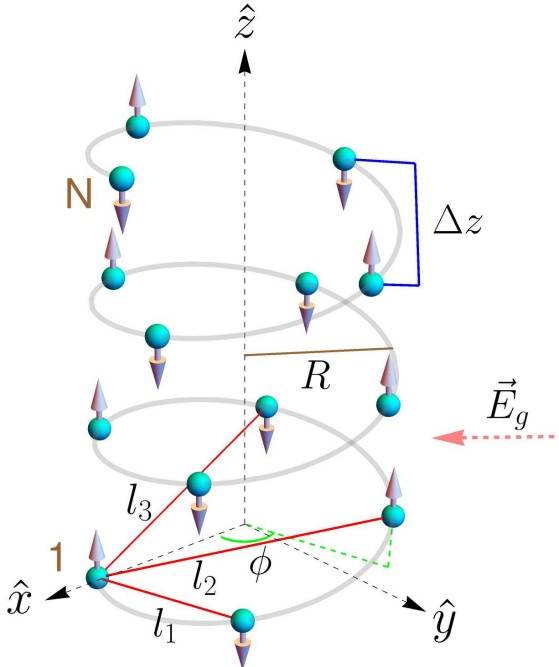

Figure 1: (Color online). Schematic diagram of an antiferromagnetic right-handed helix. Cyan balls denote the magnetic sites, where each ball represents a magnetic moment with an arrow indicating its direction. $E_g$ is the electric field, applied perpendicular to the helix axis. $l_1$, $l_2$, and $l_3$ are the first, second, and third neighbor distances, respectively. $\Delta z$ is the stacking distance, the distance along $z$-axis between two neighboring sites. $\phi = n\Delta\phi$, where $\Delta\phi$ is the twisting angle between the neighboring sites and $n$ is the site index [56].

Here $c_{n\alpha}^\dagger$ ($c_{n\alpha}$) is the creation (annihilation) operator at the $n$th site with spin $\alpha(=\uparrow,\downarrow)$. $\epsilon_n$ is the on-site potential at site $n$ and $t_m$ is the hopping integral between the sites $n$ and $n+m$.

The term $\hbar_n \cdot \boldsymbol{\sigma}$ denotes the interaction between the itenerant electron and the local moment, where $\boldsymbol{\sigma}$ is the Pauli spin vector and $\hbar_n$ is the spin-dependent scattering (SDS) parameter at site $n$. $\hbar_n = J\langle \mathbf{S}_n \rangle$ [53], where $J$ is the spin-moment exchange interaction strength and $\langle \mathbf{S}_n \rangle$ is the average spin at the $n$th site. The magnitude of the SDS parameter $|\hbar|$ is assumed to be isotropic, that is the strength is identical at each magnetic site.

In the presence of an external electric field, the site energy modifies as [44]

$$\epsilon_n = eV_g \cos(n\Delta\phi - \beta), \tag{3}$$

where $e$ is the electronic charge and $\Delta\phi$ is the twisting angle between the neighboring sites (see Fig. 1) and $V_g$ corresponds to the gate voltage associated to the applied electric field $E_g$ with $V_g = E_g \mathcal{R}$ ($\mathcal{R}$ being the radius of the helix). $\beta$ is the angle between the positive $x$-axis and the applied electric field. Such a modulation of the on-site potential (Eq. 3) can be mapped to the AAH model [11,14] with a suitable choice of $\Delta\phi$ [46] and thus a correlated disorder can be introduced into the helical system with a disorder strength $V_g$.

The second term of Eq. 1 is associated with electron hopping in different magnetic sites, where $t_m$ reads as [44,48]

$$t_m = t\, e^{-(l_m-l_1)/l_c}. \tag{4}$$

Here $l_m$ is the Euclidean distance between sites $n$ and $n+m$, $l_1$ and $l_c$ are the nearest-neighbor distance and decay constant, respectively. In terms of radius $R$ (Fig. 1) of the helix, twisting angle $\Delta\phi$, and stacking distance $\Delta z$, $l_m$ takes the form

$$l_m = \sqrt{4\mathcal{R}^2\left[\sin\left(\frac{m\Delta\phi}{2}\right)\right]^2 + [m\Delta z]^2}. \tag{5}$$

We analyze the localization behavior of the considered system using two common quantities, namely, the inverse participation ratio (IPR) and its complementary counterpart, the normalized participation ratio (NPR). For the $n$th normalized eigenstate, they are defined as [54, 55]

$$\text{IPR}_n = \sum_i |\psi_n^i|^4, \qquad \text{NPR}_n = \left(N\sum_i |\psi_n^i|^4\right)^{-1}. \tag{6}$$

In the case of a highly extended state, the IPR approaches to zero, while NPR tends to unity. Conversely, for a strongly localized state, the IPR approximately approaches to unity, and NPR goes to zero [54, 55].

To study the parameter space where localized and delocalized states coexist, $\text{IPR}_n$ and $\text{NPR}_n$ can be redefined by calculating their averages over a specified subset of states $N_L$ as [55]

$$\langle\text{IPR}\rangle = \sum_n^{N_L} \frac{\text{IPR}_n}{N_L}, \qquad \langle\text{NPR}\rangle = \sum_n^{N_L} \frac{\text{NPR}_n}{N_L}. \tag{7}$$

$\langle\text{IPR}\rangle$ and $\langle\text{NPR}\rangle$ have the same characteristic features as that of IPR and NPR, respectively. When both $\langle\text{IPR}\rangle$ and $\langle\text{NPR}\rangle$ are finite, the spatially extended and localized energy eigenstates coexist and in the corresponding parameter space one gets an SPME.

# 3 Results and discussion

For the helical system, the modified on-site energies due to the electric field can be mapped into a correlated disordered system [44–48, 56]. The effective site energy expression maps to the diagonal AAH model, where $V_g$ plays the role of AAH disorder strength. The considered orientation of magnetic moments along the $\pm z$ directions allows for the decoupling of the Hamiltonian $H$ of the helix into up and down spin sub-Hamiltonians, denoted as $H_\uparrow$ and $H_\downarrow$, respectively, that is $H = H_\uparrow + H_\downarrow$. In the absence of an electric field, $H_\uparrow$ and $H_\downarrow$ exhibit identical characteristics, leading to the absence of any spin-splitting effect. However, with the introduction of an electric field, this symmetry between the up and down spin sub-Hamiltonians is disrupted. This asymmetry arises from the distinct modification of the on-site energies experienced by up and down spin electrons under the influence of the electric field. Consequently, the previously indistinguishable behaviors of the two spin components diverge, resulting in observable spin-splitting effects within the system.

We consider a right-handed helix, characterized by specific structural parameters that render the system a short-range hopping helix analogous to a single-stranded DNA [46, 50]. The chosen parameters are radius $\mathcal{R} = 8$ Å, stacking distance $\Delta z = 4.3$ Å, twisting angle $\Delta\phi = \pi(\sqrt{5}-1)/4$, and decay constant $l_c = 0.8$ Å. The selection of $\Delta\phi$ results in on-site energies resembling an incommensurate potential, akin to the AAH disorder.

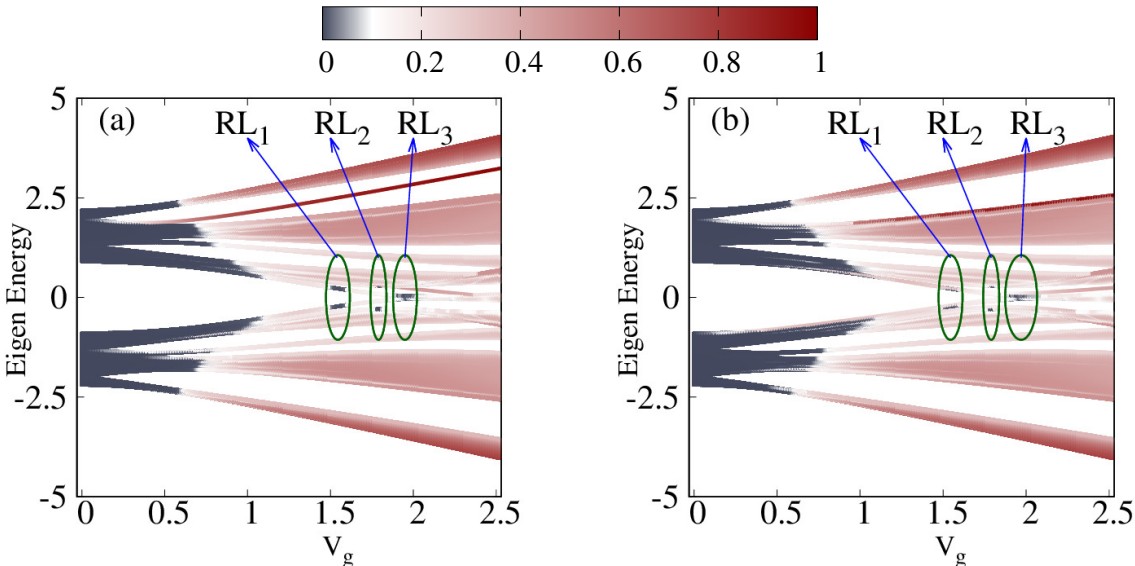

Figure 2: (Color online). Density plot. Spin-resolved IPR along with energy eigen-values as a function of gate voltage $V_g$, where (a) and (b) are associated with up and down spin electrons, respectively. Here we choose $t = 1$, $\hbar = 0.9$, $\beta = 0$, and $N = 1598$. The three instances of RL in each sub-figure are denoted with $RL_1$, $RL_2$, and $RL_3$. Their corresponding regions are marked with green ellipses.

We choose the NNH strength $t = 1\,\text{eV}$ and $\hbar = 0.9$. The direction of the electric field is assumed to be parallel to the positive $x$-axis, that is $\beta = 0$. It should be noted that the parameter $\beta$ does not have an impact on the localization properties [32].

Now, we analyze our results one by one. Let us start with the IPR characteristics of individual states (defined in Eq. 6) of the AFH in presence of transverse electric field. In Figs. 2(a) and (b), we depict the energy spectra for the up and down spin channels, respectively, showcasing the variation with the gate voltage $V_g$ (measured in units of Volts). The number of sites is taken as $N = 1598$ to make the net magnetization zero ($N$ is close to a Fibonacci number 1597). Each energy point on the plot is assigned a specific color based on its corresponding IPR value. To highlight the localization transition, the colorbar employs dark gray to represent the lowest 10% of the maximum IPR values, emphasizing extended states. The remaining portion of the color spectrum ranges from white to dark red, visually representing the increasing degree of localization. Below the threshold of approximately $V_g = 0.5$, nearly all states exhibit an extended nature, as noticed by the dark gray coloration. However, beyond $V_g \sim 1$, all states undergo a complete localization. The localization persists until around $V_g \sim 1.5$, and subsequently, we identify *three occurrences of RL* from the color-coded IPR values. The three instances of RL are highlighted by green ellipses within the approximate $V_G$-window: $RL_1$ from 1.5 to 1.6, $RL_2$ from 1.78 to 1.8, and $RL_3$ from 1.9 to 2. In the case of down spin, the energy spectrum exhibits notable differences compared to the up spin scenario, as evident in Fig. 2(b). Like the up spin case, here also we observe RL phenomenon in three different energy regions. Comparing the spectra given in Figs. 2(a) and (b), it is clearly seen that the RL regions in one spin case are shifted compared to the other. This is solely due to the breaking of symmetry between the up and down spin sub-Hamiltonians in presence of the transverse electric field. *Such a spin-specific RL phenomenon has not been addressed so far to the best of our concern*.

In the rest of our analysis, we concentrate only on the up spin case, as similar kind of behavior is expected for the down spin one.

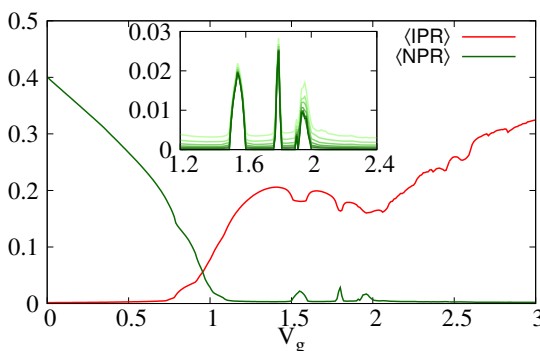

Figure 3: (Color online). $\langle IPR \rangle$ and $\langle NPR \rangle$ for the up spin electrons as a function of $V_g$ for a subset of states ranging from 30 to 70% of the eigenstates of Fig. 2. All the system parameters remain the same as described in Fig. 2. (Inset) $\langle NPR \rangle$ versus $V_g$ plot for system sizes $N = 1598, 2584, 4182, 6766, 10946, 17712$ and $N \to \infty$, represented by light to dark green color.

To observe the mixed-phase zone, we plot the behavior of $\langle IPR \rangle$ and $\langle NPR \rangle$ as a function of $V_g$ as shown in Fig. 3, represented by the the red and green colors, respectively. The averaging for $\langle IPR \rangle$ and $\langle NPR \rangle$ is performed over a subset of eigenstates, $N_L$ of Fig. 2. Specifically, the lowest 30% of the eigenstates (starting from the bottom of the spectrum) and the highest 30% are excluded in Fig. 2, focusing only on the central 40% of the spectrum. This approach ensures a more accurate and representative characterization of the localization properties in the system. The system and other parameter values remain unchanged with those in Fig. 2. In Fig. 3, both $\langle IPR \rangle$ and $\langle NPR \rangle$ become finite for $0.7 > V_g > 1.2$, indicating a critical region with a coexistence of extended and localized states. Beyond $V_g \sim 1.2$, all states become fully localized. Subsequently, both $\langle IPR \rangle$ and $\langle NPR \rangle$ attain finite values in the previously mentioned three RL regions, namely, for $1.5 > V_g > 1.6$, $1.78 > V_g > 1.8$, and $1.9 > V_g > 2$. Therefore, the system hosts as a total of four SPMEs. To mitigate potential finite-size effects, we examine the behavior of $\langle NPR \rangle$ across various system sizes, specifically, $N = 1598, 2584, 4182, 6766, 10946$, and $17712$. Using the evaluated data, we extrapolate $\langle NPR \rangle$ as $N \to \infty$. The corresponding result is presented in the inset of Fig. 3. The same subset of eigenstates as that of the main plot of Fig. 3 is used for the evaluation of $\langle NPR \rangle$. A gradient of green color, transitioning from light to dark, is employed to represent the behavior of $\langle NPR \rangle$ as a function of $V_g$ for the system sizes in ascending order. For $N \to \infty$, $\langle NPR \rangle$ attains a finite value in all the three reentrant localized regions, whereas, it converges to zero outside the RL regions. This clearly demonstrates the robustness of the occurrence of the three RLs with respect to system size and rules out any finite-size effects.

*Detail analysis of the reentrant regions*: To get a better insight about this, we characterize the localization transitions with a proper theory of phase transition following the theory of thermal phase transition. Hence, we determine the critical exponents and perform finite-size scaling analysis for the observed three regions of reentrant phases to point out the critical points and scaling behavior associated with the localization transitions.

We define an order parameter $\sigma$ to characterize the localization transition as [51, 52]

$$\sigma = \sqrt{\langle NPR \rangle} \,. \tag{8}$$

With the common notion of $\langle NPR \rangle$, $\sigma$ is also finite in extended phase and becomes zero in localized phase, which makes it suitable candidate for the order parameter for the localization phase transitions which has one-to-one correspondence with the thermal phase transition [51, 52]. There are a total of six transitions, three localized to extended phase and three

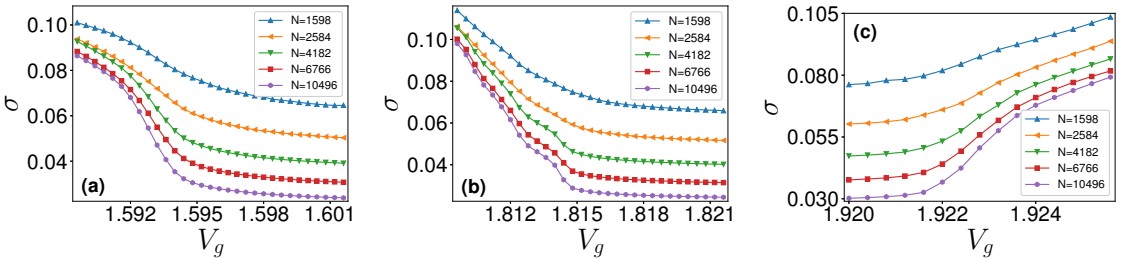

Figure 4: Plot of $\sigma$ with $V_g$ for three different transitions for system sizes $N = 1598, 2584, 4182, 6766$, and $10946$. (a) extended to localized transition in $RL_1$, (b) extended to localized transition in $RL_2$, and (c) localized to extended transition in $RL_3$.

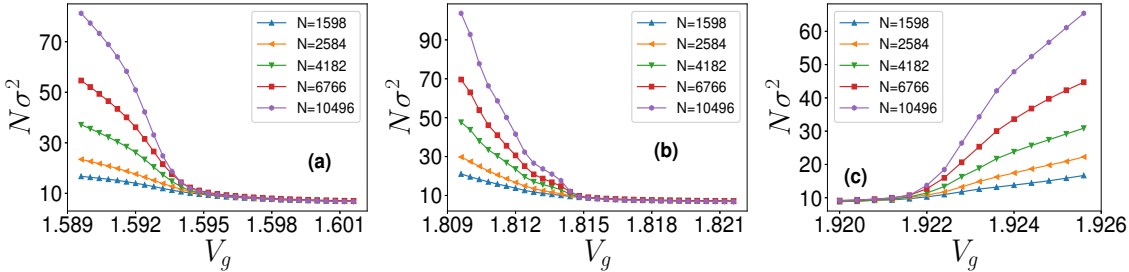

Figure 5: Variation of $\sigma^2 N$ with $V_g$ for three different transitions for system sizes $N = 1598, 2584, 4182, 6766$, and $10946$. (a) extended to localized transition in $RL_1$, (b) extended to localized transition in $RL_2$, and (c) localized to extended transition in $RL_3$.

extended to localized phase, shown in Fig. 3. Here, we choose one transition from each region for detail analysis. The variation of $\sigma$ with gate voltage $V_g$ is shown in Fig. 4(a) and Fig. 4(b), when the system moves from extended to localized phase in region $RL_1$ and $RL_2$ respectively. Whereas, the localized to extended phase variation of $\sigma$ is shown in Fig. 4(c) in the region $RL_3$. The variation of $\sigma$ with $V_g$ for all the three transitions is done for system sizes $N = 1598, 2584, 4182, 6766$, and $10496$. We find that $\sigma$ has strong finite size dependence in all three regions. With increasing the system size, $\sigma$ falls off monotonically and is almost zero for the highest system size $N = 10496$ as it should be.

Around the transition zone, the order parameter $\sigma$ varies with scaled gate voltage $\epsilon = (V_g - V_{gc})/V_{gc}$ as

$$\sigma \sim (-\epsilon)^\beta, \tag{9}$$

where $V_{gc}$ is the critical gate voltage for the transition. The participation ratio $\sigma^2 N$ [57] (fluctuation of $\sigma$) varies with scaled gate voltage $\epsilon$, across the transition zone, as

$$\sigma^2 N \sim \epsilon^{-\gamma}. \tag{10}$$

The correlation (or localization) length $\xi$ varies with scaled gate voltage $\epsilon$ in the vicinity of critical point as

$$\xi \sim |\epsilon|^{-\nu}. \tag{11}$$

Here $\beta, \gamma$, and $\nu$ are the order parameter exponent, the participation ratio exponent, and the correlation length exponent, respectively. The variation of $\sigma^2 N$ with $V_g$ for the three transitions with different system sizes are shown in Fig. 5.

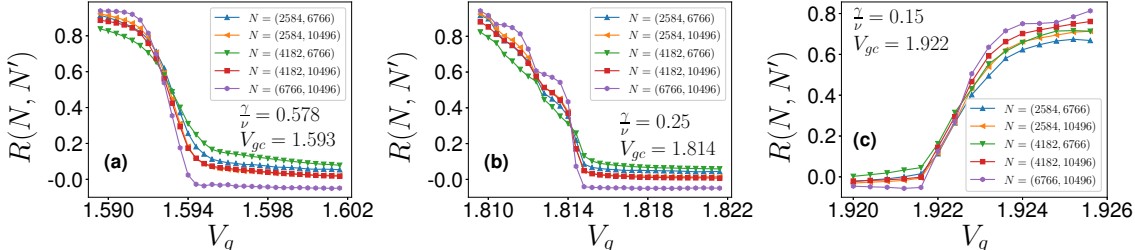

Figure 6: Plot of two system size function $R(N, N')$ with $V_g$ for three different transitions for system sizes $N = 1598, 2584, 4182, 6766,$ and $10946$ in (a) extended to localized transition in $RL_1$, (b) extended to localized transition in $RL_2$, and (c) localized to extended transition in $RL_3$.

Table 1: The critical gate voltages $V_{gc}$ and the ratios of different critical exponents $\gamma/\nu$ and $\beta/\nu$ for the three transitions.

| Transition | $V_{gc}$ | $\gamma/\nu$ | $\beta/\nu$ |
|---|---|---|---|
| $RL_1$ | 1.593 | 0.578 | 0.211 |
| $RL_2$ | 1.814 | 0.250 | 0.375 |
| $RL_3$ | 1.922 | 0.150 | 0.425 |

The critical point $V_{gc}$ and critical exponent ratio $\gamma/\nu$ for a transition can be determined using a two system size function $R(N, N')$ involving two system sizes following the prescription of Hashimoto [57].

$$R(N, N') = \frac{\ln(\sigma_N^2/\sigma_{N'}^2)}{\ln(N/N')} + 1, \tag{12}$$

where, the order parameter for system sizes $N$, and $N'$ are $\sigma_N$, and $\sigma_{N'}$, respectively. The plots of $R$ versus $V_g$ for different pairs of available $N, N'$ around the critical region intersect at a common fixed point. The abscissa of the fixed point of intersection gives the value of $V_g$, and the ordinate gives the critical exponent ratio $\gamma/\nu$ for the transition. The behavior of $R$ for the three transition regions is shown in Fig. 6. The different $R - V_g$ curves cross at the points $V_g = 1.593, 1.814,$ and $1.922$ corresponding to transitions in the regions $RL_1, RL_2,$ and $RL_3$ as noted from Figs. 6(a), (b), and (c), respectively. The values of ordinates are $R = 0.578, 0.25,$ and $0.15$, respectively for the three transitions. Hence, we have the critical gate voltages $V_{gc}$ and the critical exponent ratios $\gamma/\nu$ for the three transitions, as shown in the following Table 1. The critical exponents ratios $\beta/\nu$ can be obtained using the hyperscaling relationship [58]

$$\frac{2\beta}{\nu} + \frac{\gamma}{\nu} = d, \tag{13}$$

where $d = 1$ is the number of components of order parameter or the dimension of order parameter. The ratio between the critical exponents $\gamma/\nu$ must obey the above relationship. With $d = 1$, Eq. 13 becomes

$$\frac{\beta}{\nu} = \frac{1}{2}\left(1 - \frac{\gamma}{\nu}\right). \tag{14}$$

With the previously computed values of $\gamma/\nu$, we determine the values of critical exponents ratio $\beta/\nu = 0.211, 0.375,$ and $0.425$ for the three transitions in regions $RL_1, RL_2,$ and $RL_3$, respectively.

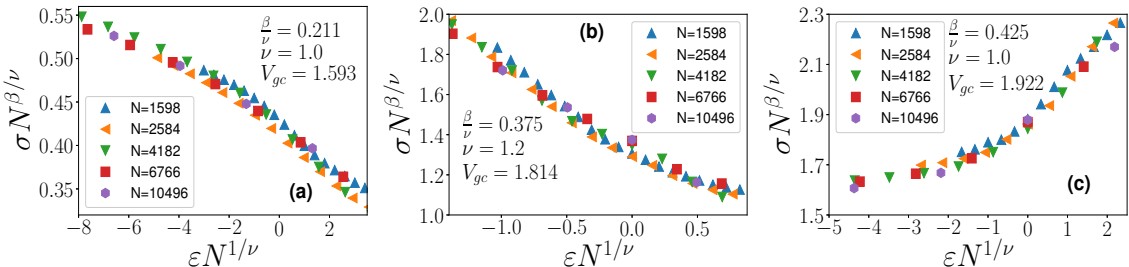

Figure 7: Plot of FSS order parameter $\sigma N^{\beta/\nu}$ with FSS gate voltage $\epsilon N^{1/\nu}$ for three different transitions for system sizes $N = 1598, 2584, 4182, 6766$, and $10946$ for (a) extended to localized transition in $RL_1$, (b) extended to localized transition in $RL_2$, and (c) localized to extended transition in $RL_3$. The FSS order parameter data for different system sizes collapses onto a single curve around the critical region for all three transitions.

*Finite size scaling*: To verify the accuracy of the computed critical exponents, we employ finite size scaling analysis. Following the theory of thermal critical phenomena [58, 59], the finite size scaling (FSS) form of $\sigma$ is assumed to have the expression

$$\sigma = N^{-\beta/\nu}\widetilde{\sigma}(\epsilon N^{1/\nu}), \tag{15}$$

where $\widetilde{\sigma}$ is a scaling functions. If the different computed critical exponents are correct, the order parameter data must collapse onto a single curve for different system sizes $N$ when the FSS order parameter $\sigma N^{\beta/\nu}$ is plotted with FSS gate voltage $\epsilon N^{1/\nu}$. In the present case, all different curves shown in Fig. 4 should fall on the same curve given by the function $\widetilde{\sigma}$ in the vicinity of critical point. The data collapse, in other words, is the verification of critical point and critical exponents. The data collapse depicted in Fig. 7(a) for extended to localized transition in $RL_1$ is obtained using $\beta/\nu = 0.211$, and $V_g = 1.593$ as obtained in Table 1. The value of $\nu$ is obtained by trial and error method. We get the best data collapse for $\nu = 1$. Similarly, we have good data collapse using the critical exponents from Table 1 for extended to localized transition in $RL_2$, and localized to extended transition in $RL_3$ shown in Figs. 7(b) and (c), respectively. The exponent $\nu$ is realized from the best data collapse for both the cases, and we get $\nu = 1.2$, and $\nu = 1$, respectively for $RL_2$, and $RL_3$. The correlation length follows the power law behavior $\xi \sim |V_g - V_{gc}|^{-\nu}$ around the critical point. Here, we see that $\xi$ varies as $|V_g - V_{gc}|^{-1}$ for $RL_1$, and $RL_3$ but $\xi \sim |V_g - V_{gc}|^{-1.2}$ for $RL_2$. It implies that the extended phase decays much faster in the case of $RL_2$ as the system deviates from $V_{gc}$, when compared to that of the other two transitions under consideration.

Similarly, the FSS form of $\sigma^2 N$ is defined as

$$\sigma^2 N = N^{\gamma/\nu}\widetilde{\chi}(\epsilon N^{1/\nu}), \tag{16}$$

where, $\widetilde{\chi}$ is another scaling function. We will get the $\sigma^2 N$ data collapse onto a single curve for different system sizes $N$ when the FSS fluctuation $\sigma^2/N^{\gamma/\nu-1}$ is plotted with FSS gate voltage $\epsilon N^{1/\nu}$. All different curves shown in Fig.(5) should fall on the same curve given by the function $\widetilde{\chi}$ in the vicinity of critical point if we use the correct critical exponents. A set of good data collapses for FSS fluctuations for all three transitions is achieved using the same set of critical exponents used in case of the data collapse of FSS order parameter given in Table 1. The data collapses validate the critical exponents tabulated in Table 1. Figure 8 depicts the data collapse of FSS fluctuation for all three different transitions. The $\nu$ exponents are also identical with the previous cases, that is, $\nu = 1.0, 1.2$, and $1.0$ for $RL_1, RL_2$, and $RL_3$, respectively.

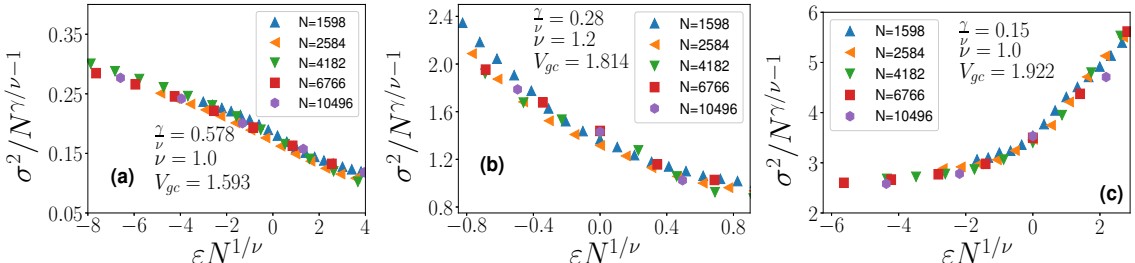

Figure 8: Plot of FSS fluctuation $\sigma^2/N^{\gamma/\nu-1}$ with FSS gate voltage $\epsilon N^{1/\nu}$ for three different transitions for system sizes $N = 1598, 2584, 4182, 6766$, and $10946$ for (a) extended to localized transition in $\mathrm{RL}_1$, (b) extended to localized transition in $\mathrm{RL}_2$, and (c) localized to extended transition in $\mathrm{RL}_3$. The FSS $\sigma^2/N$ data for different system sizes collapses onto a single curve around the critical region for all three transitions.

The critical exponents $\beta/\nu$ and $\gamma/\nu$ can be directly extracted from the finite-size dependent quantities $\sigma$ and $\sigma^2 N$ at the critical gate voltage $V_{gc}$, or equivalently, at $\epsilon = 0$. Equation 15 at the critical gate voltage $\epsilon = 0$ becomes

$$\sigma_N = \sigma(\epsilon = 0) = N^{-\beta/\nu}\widetilde{\sigma}(0), \qquad (17)$$

where $\widetilde{\sigma}(0)$ is a constant. Thus, system size dependent order parameter $\sigma_N$ becomes a function of $N$. Taking logarithm,

$$\ln(\sigma_N) = -\frac{\beta}{\nu}\ln(N) + \ln(\widetilde{\sigma}(0)). \qquad (18)$$

Hence, a plot of critical $\ln(\sigma_N)$ versus $\ln(N)$ should give a straight line with a slope $-\frac{\beta}{\nu}$. The critical values of $\ln(\sigma_N)$ is plotted with $\ln(N)$ for system sizes $N = 1598, 2584, 4182, 6766$, and $10946$ for the extended to localized transition in $\mathrm{RL}_1$, extended to localized transition in $\mathrm{RL}_2$, and localized to extended transition in $\mathrm{RL}_3$ in Fig. 9(a). The slopes or the exponent ratios $\frac{\beta}{\nu}$ are found to be $0.1892 \pm 0.134, 0.3509 \pm 0.11$, and $0.4247 \pm 0.0063$, which are very close to the critical exponents tabulated in the Table 1.

Equation 16 at the critical gate voltage $\epsilon = 0$ becomes

$$\sigma^2 N = \sigma_N^2 N(\epsilon = 0) = N^{\gamma/\nu}\widetilde{\chi}(0), \qquad (19)$$

where $\widetilde{\chi}(0)$ is a constant. Thus, system size dependent order parameter fluctuation $N\sigma_N^2$ becomes a function of $N$. Taking logarithm,

$$\ln(N\sigma_N^2) = -\frac{\gamma}{\nu}\ln(N) + \ln(\widetilde{\chi}(0)). \qquad (20)$$

Hence, a plot of critical $\ln(N\sigma_N^2)$ versus $\ln(N)$ should give a straight line with a slope $\frac{\gamma}{\nu}$. The critical values of $\ln(N\sigma_N^2)$ is plotted with $\ln(N)$ for system sizes $N = 1598, 2584, 4182, 6766$, and $10946$ for the three different transitions in $\mathrm{RL}_1$, $\mathrm{RL}_2$, and $\mathrm{RL}_3$ in Fig.(9b). The slope $\frac{\gamma}{\nu}$ is found to be $0.6217 \pm 0.0268, 0.2982 \pm 0.022$, and $0.1505 \pm 0.0126$ for $\mathrm{RL}_1$, $\mathrm{RL}_2$, and $\mathrm{RL}_3$, respectively, which are again very close to the critical exponents tabulated in the Table 1.

*Parameter space*: With the confidence that the three RLs are not due to any finite-size effect, we explore the parameter space to identify the conditions under which these three RLs exist. To do so, we compute $\eta$, defined as [60]

$$\eta = \log_{10}\left[\langle\mathrm{IPR}\rangle \times \langle\mathrm{NPR}\rangle\right]. \qquad (21)$$

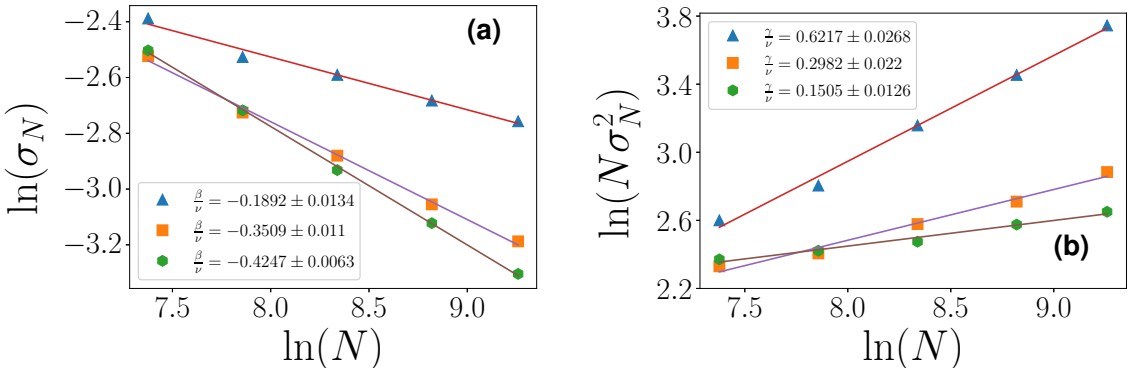

Figure 9: Direct determination of critical exponent ratio. (a) The critical values of $\ln(\sigma_N)$ is plotted with $\ln(N)$ for system sizes $N = 1598, 2584, 4182, 6766$, and $10946$ for the extended to localized transition in $RL_1$, extended to localized transition in $RL_2$, and localized to extended transition in $RL_3$. The critical exponent ratio $\beta/\nu$ is determined from the slopes. (b) The critical values of $\ln(N\sigma_N^2)$ is plotted with $\ln(N)$ for system sizes $N = 1598, 2584, 4182, 6766$, and $10946$ for the three different transitions in $RL_1$, $RL_2$, and $RL_3$. The critical exponent ratio $\gamma/\nu$ is obtained from the slopes.

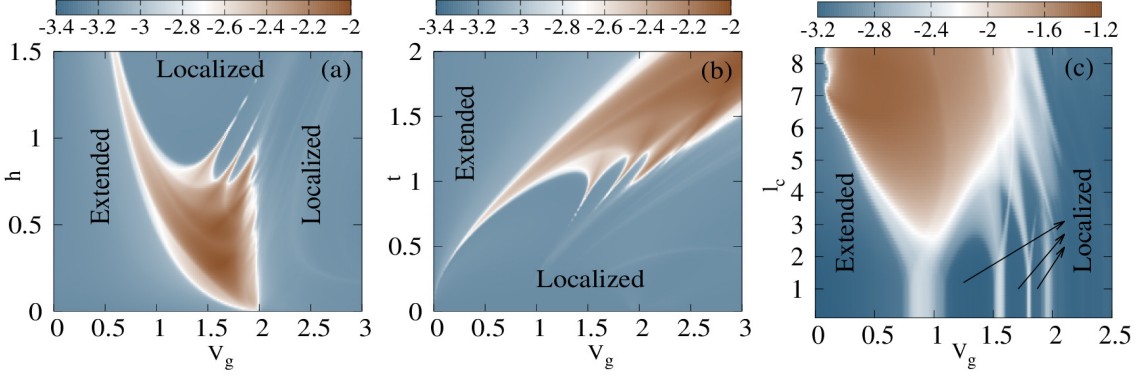

Figure 10: (Color online). Density plot of $\eta$ for the up spin case in the (a) $V_g$-$h$, (b) $V_g$-$t$, and (c) $V_g$-$l_c$ planes. The system size and all the other parameters remain the same as described in Fig. 2.

In the mixed phase zone, both $\langle IPR \rangle$ and $\langle NPR \rangle$ are finite and $\mathcal{O}(1)$, yielding $\eta$ within $-2 \leq \eta \leq -1$. Conversely, in localized (extended) regime, $\langle NPR \rangle$ ($\langle IPR \rangle$) tends toward $\sim N^{-1}$, and $\eta < -\log_{10} N$. For instance, at $N \sim 10^3$, $\eta < -3$. Hence, the quantity $\eta$ serves as a clear discriminator between fully extended or localized phase and a mixed phase.

First, we explore the behavior of $\eta$ in the phase space of $V_g$ and $h$ for the up spin channel as shown in Fig. 10(a). All other parameters remain constant as indicated in Fig. 2. In the colorbar of $\eta$, steel blue color corresponds to the extended or localized phase, while the brown color represents the mixed phase. In Fig. 10(a), the localized or extended phases are determined by analyzing the $\langle IPR \rangle$ and $\langle NPR \rangle$ values. All three RLs emerge within the $h$-range of approximately 0.75 to 0.96. The third RL ceases to exist beyond $h \sim 0.96$. Additionally, no instances of RL are present beyond $h \sim 1.2$. This observation strongly suggests that RL occurrence is possible when $h$ is comparable to the numerical value of $V_g$.

Second, we study the $\eta$-behavior in the phase space of $V_g$ and $t$ for the up spin channel as shown in Fig. 10(b). All other parameters kept unchanged as indicated in Fig. 2. As both $V_g$ and $t$ increase from zero to a finite value, the width of the first critical region expands. The first RL emerges around $t \sim 0.8$. The second RL becomes noticeable around $t \sim 0.9$, and the third one around $t \sim 1$. The first RL diminishes beyond $t \sim 1.1$, the second one around $t \sim 1.15$, and the third one for $t \sim 1.25$. All three RLs are pronounced within the range of approximately $1 < t < 1.1$. This range is comparable to the numerical value of $\hbar$, which is fixed at $\hbar = 0.9$, consistent with the previous analysis. After $t \sim 1.25$, all RLs vanish and merge into the first critical region.

So far, all the results were discussed for the scenario of short-range hopping. To investigate the transition from short-range hopping to long-range hopping and understand the localization behavior, we examine the $\eta$-behavior within the $V_g$ and $l_c$ phase space, as illustrated in Fig 10(c). The variation of the decay constant spans from 0.1 Å to 8.5 Å, thereby inducing a shift from a short-range hopping regime to a long-range hopping scenario. In accordance with Fig. 2, the remaining parameters are maintained at their specified values without any alterations. For low values of $l_c$ (primarily SRH) within the range of 0.1 to 2 Å, the $V_g$-window associated with the first critical region remains nearly constant and it expands as $l_c$ increases. The three RLs are present right from $l_c = 0.1$ Å. The first RL converges with the first critical region at approximately $l_c \sim 0.75$ Å, but the second and third RLs persist with increasing $l_c$. The second RL unites with the first critical region while the third one survives till $l_c \sim 4$ Å. Ultimately, the third RL disappears at approximately $l_c \sim 5$ Å, leaving only one critical region beyond that point. Another notable observation is that from the first RL, a secondary RL emerges within a short $V_g$-window with $l_c$ between 3 to 4 Å, and subsequently integrates with the second RL. A similar scenario is observed for the second RL, wherein another secondary RL originates from $l_c \sim 2$ Å and then dissipates into the localized region beyond $l_c \sim 3$ Å.

*Experimental possibilities of antiferromagnetic helix*: AFH structures have been successfully fabricated by several research groups [61–67]. For instance, a metallic spiral antiferromagnetic system [61] has been realized in $SrFeO_{2.95}$, long-range helical antiferromagnetic ordering [62] has been observed in polycrystalline samples of $Lu_{1-x}Sc_xMnSi$, and incommensurate antiferromagnetic spiral-like structures [63, 64] have been reported in $EuNi_2As_2$ and $EuCo_2As_2$. Additional examples include spin-canted antiferromagnetism with helical topology [65], canted antiferromagnetic ordering [66], and the potential realization of curvilinear one-dimensional antiferromagnets [67].

It is important to highlight that these studies predominantly utilize heavy magnetic elements, which can also be theoretically described by the Hamiltonian used in the present work. Similar Hamiltonians have been employed successfully in previous studies, such as those by Takahashi and Igarashi [68] for $La_2CuO_4$ and $Sr_2CuO_2Cl_2$, as well as by others [69, 70]. Moreover, in the Hamiltonian of our present work, the itinerant electrons interact with localized magnetic moments at different lattice sites resulting in a spin-dependent phenomena and the interactions between the neighboring magnetic moments have been ignored. The moment-moment interaction does not essentially yield any such new localization behavior as this interaction term can be written as a sum of two terms, under mean-field approximation, where one term is associated with Zeeman like interaction and the other term is a constant one. Though the Zeeman interaction provides a spin-dependent scattering, it is extremely weak compared to the spin-moment scattering what we have considered in our Hamiltonian. Thus, in light of the aforementioned studies, the use of the Hamiltonian in describing antiferromagnetic helices is well justified, suggesting that the RL behavior may consequently be observed.

Furthermore, the entire analysis was conducted at zero temperature. While non-zero temperatures could also have been considered, at low temperatures, quantum fluctuations would arise, potentially altering the orientation of magnetic moments. However, these fluctuations

are unlikely to significantly impact the localization phenomena studied here, as the strong spin-moment scattering induces a substantial mismatch between the up- and down-spin energy channels. A detailed investigation of finite-temperature effects, including quantum fluctuations, could be pursued in future work to provide deeper insights into the system.

Finally, the question remains whether such RL phenomena can be observed experimentally and whether the necessary facilities exist to explore them. To the best of our knowledge, probably no experimental references are currently available regarding RL phenomena in an antiferromagnetic helix under a transverse electric field. However, as noted earlier, similar systems have already been established in the literature, suggesting that our results are experimentally verifiable. We hope that experimental studies addressing RL phenomena will emerge soon.

While our work is theoretical, we propose a potential and relatively straightforward experiment to observe RL behavior by measuring the junction current as a function of applied bias voltage at varying disorder strengths, where, the disorder strength can be varied by the applied transverse electric field. In the weak disorder regime, where states are predominantly extended, the current is expected to increase with bias. Conversely, in the localized regime with higher disorder, the current should decrease. In the reentrant regimes, when some states become delocalized, an increase in current may be observed. Thus, RL behavior can be effectively studied through current measurements. Nevertheless, experimental experts may devise more refined approaches for this purpose.

# 4   Summary

The present investigation has revealed the occurrence of multiple spin-dependent reentrant localization in a helical system when subjected to an electric field with net zero magnetization, characterized by antiferromagnetic ordering of the moments. Crucially, this accomplishment has been realized without incorporating any hopping dimerization scenario, a factor upon which previous studies attributing the occurrence of RL had relied. Our study of spin-resolved IPR values across various energy states has revealed the presence of three distinct RLs. The robustness of these three RLs has been affirmed through a comprehensive analysis of averaged IPR and NPR values in the limit as the system size approaches infinity ($N \to \infty$).

To validate the observed transitions, we have defined the order parameter and its fluctuations for localization phase transitions in direct analogy to thermal phase transitions. The critical exponent ratios $\gamma/\nu$ and $\beta/\nu$ have determined from the two-system size function $R(N, N')$ and a hyperscaling relationship, respectively. The $\nu$ exponent has been obtained by trial and error method, ensuring the best data collapse of the FSS variables. We have validated the critical regions and the measured critical exponents through data collapse, with satisfactory results obtained by plotting FSS variables and measured critical exponents across the three critical regions. We have also extracted the critical exponents at the critical point directly from the system size-dependent order parameters and order parameter fluctuations. The values of the critical exponents obtained by both methods are in good agreement. The extended phases decay much faster during the localization transition in region RL2 as the system deviates from $V_{gc}$ compared to the other two localization transitions.

The investigation of $\eta$-behavior unravels that RL phenomenon becomes apparent when the electric field strength $V_g$ is on a comparable scale to the value of $\hbar$. Additionally, in the $V_g$-$l_c$ plane, RLs vanish as the system undergoes from the short-range hopping to the long-range hopping case. Furthermore, our investigation has identified two secondary RLs stemming from the first and second RLs, which however warrants further in-depth exploration.

Before we end, we would like to point out that the practical implications of spin-dependent reentrant localization phenomenon in the absence of any hopping dimerization may provide innovative applications in quantum technologies. By suitably defining the spin qubits, spin-dependent RL phenomena may find applications in quantum computing, where stable qubits are essential for reliable gate operations and entanglement. Additionally, these qubits could be applied in spintronic devices for spin-based information storage and processing, exploiting the tunability of the localization transitions. The ability to control spin-dependent behavior in systems exhibiting reentrant localization also opens avenues for quantum sensing and quantum communication, where precise manipulation of quantum states is crucial for robust information transfer and detection. The investigation of the RL behavior in the $V_g$-$l_c$ plane might help to understand and to control localization phenomena in similar kind of other fascinating helical systems. With other types of antiferromagnetic ordering, such as non-colinear or non-coplaner structures, different critical points may emerge for different spin species in these systems.

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
