# Peer review of "Critical analysis of multiple reentrant localization in an antiferromagnetic helix with transverse electric field: Hopping dimerization-free scenario"

_SciPost Physics, doi:SciPost Phys. Core 8, 012 (2025)_

## Round 3 · Referee Report · Anonymous (Referee 1) · 2024-11-7

Strengths

1 - The results are presented in a clear way.
2 - The storyline and the messages are clear and polished.
3 - The scaling analysis has been preformed thoroughly.

Weaknesses

1 - Some obvious things are presented as big puzzles and important questions.
2 - The motivation behind introducing the particular model as well as its potential realisation in experiments are severely lacking.
3 - The introduction is poorly written and contains several important mistakes.
4 - The concluding remarks regarding the importance of this work are too vague, without a clear explanation why they should be true.

Report

I think this paper deserves to be published in SciPost after my comments below have been properly addressed.

Requested changes

1 - First sentence of the introduction is incorrect. Not all systems show M-I transition in the presence of disorder. Be more specific with this claim, i.e. refer to uncorrelated 1D lattice models only.

2 - Second paragraph of the introduction starts with: “In contrast to the uncorrelated disorder, correlated disorder provides the advantage of a sharply-defined critical point for the extended-localized phase transition”. Again, this does not hold for all correlated potentials. Make the sentence more specific to the models where the claim is necessary true.

3 - The first mention of the reentrant localisation transition has been done by Hiramoto and Kohmoto, so please cite their work Phys. Rev. Lett. 62, 2714 (1989).

4 - The end of the third paragraph of the introduction has to be rewritten slightly. This is simply because the answer to the question posed by there: “is it possible to implement a system where the RL phenomenon can be obtained in the ‘absence’ of hopping dimerization?” is YES, just look at the so called interpolating Aubry-Andre-Fibonacci model in Refs.[31,32] as well as Phys. Rev. Lett. 62, 2714 (1989) and arXiv:2406.14193.

5 - In the beginning of the third paragraph in “SYSTEM AND THEORETICAL FRAMEWORK”, the authors mention “incoming electron” without ever properly referring to it afterwards. Where is that electron coming from?

6 - Define $\Delta \phi$ earlier, i.e. immediately after Eq.(3).

7 - How is the subset of states $N_L$ defined in Eq.(7) and afterwards. More precisely, how are those states chosen in practice?

8 - In the second and third paragraph of section “RESULTS AND DISCUSSION”, the authors put some values for their parameters. Please motivate those values. Are they taken from some experiments, or are they chosen arbitrarily?

9 - In the text, the authors refer to the phase where the mobility edge is present as “critical phase”, “critical regime” and “mixed phase zone”, if I understood it correctly. This is confusing. Firstly, did I understood it correctly that the three phrases describe the same thing in the text. And secondly, why is the region with a mobility edge critical (there are no critical states there)?

10 - The authors work in a zero-magnetisation sector of the spin chain, assuming that the AFM coupling is the largest energy scale of the problem. I'm wondering if this is a realistic assumption? In the realistic spin chains, the quantum fluctuations and the ones due to finite temperature are inevitable. How would the spin-flips (i.e. other magnetisation sectors) caused by fluctuations change the localisation properties discussed in this work?

11- In the last paragraph of the paper, the authors claim “... may provide innovative applications in quantum technologies”. I'm asking them to either elaborate more on this and show which applications in quantum technologies can this work influence and how, or to remove the sentence.

12 - Similar to my previous comment, in the next sentence, the authors claim “...to control localization phenomena in similar kind of other fascinating helical systems.”. Since there is no discussion about a particular physical system that is described by this model, I do not see how one can “control” the localisation phenomena in “other fascinating helical systems”. I ask the authors to add a few paragraphs about exact physical systems that are captured by the model discussed in this work. Furthermore, it would be nice to discuss and cite some existing or potential experiments that can observe such fine-tuned localisation properties.

Recommendation

Ask for minor revision

---

## Round 4 · Referee Report · Anonymous (Referee 1) · 2024-12-4

Report

The authors successfully answered questions from my last report and incorporated all required changes. Therefore, I recommend the publication of the manuscript in its current form.

I think the manuscript deserves to be published in SciPost Physics Core rather than SciPost Physics since I would disagree that it
“opens a new pathway in the research” or that the investigated problem has been “a long-standing research stumbling block” in the current research on localization phenomena in these types of systems.

Recommendation

Publish (meets expectations and criteria for this Journal)

---

## Round 4 · List of Changes

Attached.

---

## Editorial Decision

published